# A Greater Adherence to the Mediterranean Diet Supplemented with Extra Virgin Olive Oil and Nuts During Pregnancy Is Associated with Improved Offspring Health at Six Years of Age

**DOI:** 10.3390/nu17101719

**Published:** 2025-05-19

**Authors:** Laura del Valle, Verónica Melero, Andrés Bodas, Rocío Martín O’Connor, Ana Ramos-Levi, Ana Barabash, Johanna Valerio, Paz de Miguel, José Ángel Díaz Pérez, Cristina Familiar Casado, Inmaculada Moraga Guerrero, Inés Jiménez-Varas, Clara Marcuello Foncillas, Mario Pazos, Miguel A. Rubio-Herrera, Bricia López-Plaza, Isabelle Runkle, Pilar Matía-Martín, Alfonso L. Calle-Pascual

**Affiliations:** 1Endocrinology and Nutrition Department, Hospital Clínico Universitario San Carlos, Instituto de Investigación Sanitaria San Carlos (IdISSC), 28040 Madrid, Spain; lauradel_valle@hotmail.com (L.d.V.); veronica.meleroalvarez10@gmail.com (V.M.); rmoconnor@salud.madrid.org (R.M.O.); ana_ramoslevi@hotmail.com (A.R.-L.); ana.barabash@gmail.com (A.B.); valeriojohanna@gmail.com (J.V.); pazdemiguelnovoa@gmail.com (P.d.M.); joseangel.diaz@salud.madrid.org (J.Á.D.P.); cristinafamiliarcasado@gmail.com (C.F.C.); inmamgg@hotmail.com (I.M.G.); i.jimenez.varas@gmail.com (I.J.-V.); clara994@hotmail.com (C.M.F.); mario.pazos@salud.madrid.org (M.P.); marubioh@gmail.com (M.A.R.-H.); irunkledelavega@gmail.com (I.R.); 2Paediatrics Department, Hospital Clínico Universitario San Carlos, Instituto de Investigación Sanitaria San Carlos (IdISSC), 28040 Madrid, Spain; abodas@pdi.ucm.es; 3Facultad de Medicina, Medicina II Department, Universidad Complutense de Madrid, 28040 Madrid, Spain; blopezpl@ucm.es; 4Centro de Investigación Biomédica en Red de Diabetes y Enfermedades Metabólicas Asociadas (CIBERDEM), 28029 Madrid, Spain

**Keywords:** Mediterranean diet, extra virgin olive oil, pistachios, pregnancy, children’s health, atopic eczema, bronchiolitis

## Abstract

**Background/Objectives:** Children’s health may be influenced by maternal eating habits during pregnancy. A Mediterranean diet (MedDiet)-based pattern has been associated with benefits in infectious disease and allergies in children. **Methods:** From a total 2228 pregnant women evaluated between 2015 and 2017 in the St. Carlos cohort, 1292 women belonging to the intervention group (IG) undertook a MedDiet supplemented with extra virgin olive oil (EVOO) and nuts from early on in pregnancy. The control group (CG) consisted of 516 pregnant women who were advised to follow a fat-restricted MedDiet. The modified 12-point Mediterranean diet adherence screener (MEDAS) was applied. A total of 1808 (81.2%) children were analyzed at 6 years postpartum. All women from both groups received the same postpartum nutritional recommendations as the IG had during pregnancy. **Results:** Children from the IG showed lower rates of bronchiolitis and atopic eczema compared to the CG. Children whose mother’s MEDAS score was ≥6 at 24–28 GW vs. MEDAS scores < 6 presented lower rates of bronchiolitis [222/695 (28.8%) vs. 441/1113 (39.6%); *p* = 0.022] and atopic eczema [311/695 (39.0%) vs. 564/1113 (50.7%); *p* = 0.039]. The offspring of mothers with MEDAS scores ≥ 8 (*n* = 176) vs. MEDAS scores ≤ 3 (*n* = 564) showed a lower relative risk (RR) (95% CI) for developing bronchiolitis [0.75 (0.58–0.97)] and atopic eczema [0.82 (0.69–0.98)], with both having a *p* < 0.001 trend. **Conclusions:** A higher adherence to a MedDiet supplemented with EVOO and nuts during pregnancy is associated with health benefits in offspring at 6 years of age.

## 1. Introduction

Over the last few years, there has been increasing interest in the influence of maternal nutrition during pregnancy on the development of non-communicable diseases in offspring. In fact, a woman’s diet during pregnancy exerts an important effect on the intra-uterine environment, influencing fetal growth and development [1,2,3].

However, evidence suggests that the maternal metabolic status could also exert effects on the incidence of metabolic, allergic, and inflammatory diseases in childhood [4,5,6,7]. That nutrition can induce phenotypic modifications in the fetus has been recognized since the Agouti factor model was first recognized in the middle of the last century, through a mechanism we now term “epigenetic” [8,9,10,11,12,13].

A maternal Mediterranean diet (MedDiet) has been shown to reduce perinatal complications and prevent allergic and infectious diseases in offspring [3,4,14,15,16].

A prior analysis conducted by our group (the St. Carlos gestational diabetes mellitus (GDM) prevention studies cohort) detected an association between a MedDiet-based nutritional intervention early in pregnancy and a decrease in the rates of hospital admissions in 2-year-old children diagnosed with bronchiolitis/asthma or any adverse event requiring antibiotics and/or corticosteroids, but differences in the rates/prevalence of these diseases were not found [17]. However, the influence of breastfeeding, the adherence to the mandatory vaccination schedule, and early socialization in daycare centers may mask the influence of nutrition during pregnancy on the health of the offspring [18,19,20].

The aim of the present study is to evaluate the health status of the offspring of the St. Carlos GDM prevention cohort at 6 years postpartum. We hypothesized that a MedDiet-based nutritional intervention in pregnancy, supplemented with extra virgin olive oil (EVOO) and nuts, is associated with improved health outcomes in children at 6 years of age, compared to those whose mothers followed a similar diet without the aforementioned supplementation.

## 2. Research Design and Methods

### 2.1. Study Design

This is a prospective, unicentric, interventional analysis of the “St. Carlos GDM prevention Study”. The registered trials were conducted under the following codes: “ISRCTN84389045, ISRCTN13389832, and ISRCTN16896947”. They were approved by the Ethics Committee of the Hospital Clínico San Carlos (ethical codes: “CI 13/296-E, CI 16/442-E, and CI 16/316”, respectively). The study was conducted following the principles of the Declaration of Helsinki. The three trials were consecutive.

The St. Carlos cohort comprises the participants of three clinical trials evaluating the effect of a MedDiet-based intervention on pregnant women, initiated before the 12th gestational week (GW). The specifics of the interventions in the three groups have been previously described. Briefly, the first study (ISRCTN84389045), starting in 2015, was a randomized controlled trial (RCT) in which women were allocated into an intervention group (IG) and a control group (CG). Both groups were instructed to eat a MedDiet. But women in the IG were told to increase the consumption of EVOO to over 40 mL/day and to eat a handful of pistachios daily, with both provided free of cost. The CG women, however, were urged to limit fat intake. The second trial was a real-world clinical study (ISRCTN13389832) in which all participants continued with the same dietary recommendations as the IG of the first trial, but without the provision of EVOO and pistachios. The third trial (ISRCTN16896947) was also a RCT in which women with a BMI ≥ 25 kg/m^2^ were included. In this case, the participants belonging to the IG were urged to increase the consumption of EVOO and nuts, but, in contrast to the prior studies, only pistachios were provided at no cost.

The current study analyzes the children of women who participated in these prior clinical trials at 6 years of age, coinciding with mandatory Spanish pediatric monitoring. At the onset of the study, 2529 normoglycemic pregnant women were included before the 12th GW, 2228 of whom were followed until delivery.

During the first prenatal (PN) visit (10–12 GW), information about age, ethnicity, personal and family medical history, obstetric history (GDM/miscarriages), employment, and educational status was collected. Also registered were data on gestational age as estimated by the first ultrasound, declared pre-pregnancy body weight (BW), BMI (kg/m^2^), blood pressure (mmHg) while seated after rest (BP), and fasting serum glucose (FSG) levels (mg/dL). Throughout the pregnancy, two additional visits were conducted: one between 24 and 28 GW, and another at 36–38 GW and/or delivery hospital discharge.

Subsequently, a visit was made in the immediate postnatal (Pn) period, at three months, during which a nutritional intervention based on MedDiet recommendations was offered. All women received the same Pn period nutritional recommendations as the IG had during pregnancy. During this visit, perinatal data were also collected, including gestational age at birth, type of delivery, Apgar score, umbilical cord pH, and newborn birth weight length and centiles. Clinical measures were updated and recorded in the visits at 24–28 GW, 36–38 GW, and Pn, including BW, BMI, BP at rest, and FSG. Later, participants were invited to attend a follow-up visit at 6 years Pn period.

During in-office outpatient appointments, the lifestyle and eating habits of mothers were evaluated through two questionnaires. The first was a modified version of the Mediterranean Diet Adherence Screener (MEDAS) [21], adapted to exclude alcohol and juices, as their consumption is not recommended during pregnancy. This questionnaire consists of 12 items, with a total score ranging from 0 to 12, where higher scores indicate greater adherence to the Mediterranean dietary pattern. The second questionnaire used was the Diabetes Nutrition and Complications Trial (DNCT) [22], which assesses the frequency of food consumption and physical activity. Both questionnaires can be found in previously published papers [14,17].

### 2.2. Offspring Cohorts and Children’s Data Collection

The flowchart can be seen in Figure 1.

At 6-year Pn follow-up, a total of 1808 children were studied. In total, 516 (28.53%) had mothers who had been in the CG (CGC), and 1292 had mothers from the IG (IGC).

Children’s information was obtained both from the mother’s face-to-face visit, as well as from the electronic records of the health care system. The Community of Madrid provides universal pediatric care, free at point-of-use, through the SERMAS, the organization responsible for the public health care system in Madrid. The electronic records (HORUS) include hospital discharge and emergency room reports, as well as out-patient visits, both at the primary care and hospital/specialist level. Thus, a complete and up-to-date view of the child’s clinical history is available. Additionally, mothers whose children were also seen in the private sector were asked to provide all relevant information. All data were recorded at the 6-year visit.

Information regarding breastfeeding was collected and included the duration of both exclusive and mixed breastfeeding. All data provided by the mothers during the in-office visit were cross-referenced using the pediatrician’s records from HORUS. Additionally, the age of introduction of gluten-free and whole cereal was registered, as was attendance to daycare (yes/no) and the age (in months) at which the child started.

The vaccine calendar (mandatory and recommended vaccines) was obtained from the pediatric health register. By the age of 6, most vaccines had been administered to complete the vaccination schedule and immunization. Additionally, some recommended vaccines (Meningococcal C, ACWY, and COVID-19) had been included at this stage. Vaccination was considered complete when children had received all doses per the official Spanish vaccination calendar [23]. COVID-19 diagnosis was confirmed either through pediatric records or test results registered in HORUS. In cases where no positive test result was confirmed, the disease was considered not acquired.

Information on the number of admissions, the type of hospital admission (medical/surgical and the hospital department involved), and hospital discharge reports was registered.

Detailed data on pharmacological treatments were collected, including the number of episodes of the treated disease and the duration of therapy. Recorded information included the use of antibiotics, corticosteroids, and bronchodilators, with information on their administration (yes/no), frequency of use, and reasons for administration. Additionally, the use of antihistamines was recorded, although only information on the need for use was collected (yes/no).

At the mandatory Madrid pediatric visit for 6-year-olds, children’s age, BW, height, and BMI were recorded. *Z*-scores were calculated based on these measurements to assess growth and anthropometric development [24].

Registered food allergies included a diagnosis of cow’s milk protein, gluten, eggs, nuts, fruits and/or vegetables, legumes, and fish and/or shellfish. Additional allergies, such as to antibiotics, sun exposure, insect bites, animal hair, and seasonal allergies, were also recorded.

The diagnoses of asthma, atopic eczema, and bronchiolitis (total number of episodes and the age of the first episode) were recorded based on whether the children had been diagnosed with these conditions and if they continued with these diagnoses at the age of 6.

Information on development and neurological and psychomotor maturation were also registered. Mothers were asked about the children’s bladder control: “At what age did your child consistently demonstrate daytime bladder control, request use of the toilet, and keep their diaper dry for several consecutive days?” Similarly, questions were asked about the age at which daytime anal sphincter control and night-time bladder control were achieved. Considering neurocognitive maturation, women were asked the following question: “At what age did your child begin using the index finger to point at objects or people with communicative intention?” This information would allow an analysis of the relationship between pointing and other developmental milestones. To assess locomotor development, mothers were asked the following: “At what age did your child begin to stand independently, and take his/her first steps?”

Furthermore, The International Study of Asthma and Allergies in Childhood (ISAAC) [25] questionnaire was applied. This is a standardized assessment tool designed to collect information on the respiratory and allergic health of children. The questionnaire evaluates the prevalence and characteristics of asthma and allergies in childhood. In this study, the ISAAC questionnaire was completed by the mothers of the analyzed offspring during the face-to-face interview, with health personnel available to respond to any doubts the women could have.

### 2.3. Statistical Analysis

Statistical analyses were performed using SPSS 15.0. Continuous variables are expressed as mean, and SD. Categorical data are expressed as number and percentage. The ANOVA was used for trend and nonparametric Mann–Whitney and Kruskal–Wallis tests were used to evaluate differences between continuous variables with no normal distribution. The chi-square test was used to assess differences between categorical variables. Binary logistic study was performed and MEDAS scores at week 24–28 GW were stratified as independent values. Values below or above mean value 5.8 (as ≥6 and <6) and the score/case association were analyzed considering the score 0–3 as a reference group and the relative risk (RR), and the 95% confidence interval (95% CI) was assessed by scores between 4–5, 6–7, and 8–12, respectively. A *p* value of <0.05 was considered significant.

## 3. Results

From the 2228 women studied throughout pregnancy and delivery, 516 children from the CG (CGC) and 1292 children from the IG (IGC) were analyzed at the 6-year follow-up.

Women in the IG had higher MEDAS and nutrition scores throughout the second and third trimesters. These results correlated with lower levels of FSG and rates of GDM diagnosis in both trimesters. Table 1 shows the data collected during the pregnancy of the mothers of the children evaluated in this analysis.

Characteristics of mothers during gestation, delivery and newborn data are displayed in Appendix A.

Despite breastfeeding, the introduction of cereals and the vaccination schedules were similar between groups. The percentage of children attending daycare was higher in the IGC, as was the age at the onset of attendance. IGC were significantly earlier walkers as compared to CGC children. Rates of bronchiolitis and atopic dermatitis were significantly lower in IGC. Furthermore, children from IGC tended to require pharmacological therapy less frequently than CGC and tended to have shorter hospital length-of-stay (LOS) when admitted to hospital. Table 2 shows children’s data according to their mothers’ randomization to IG or CG

Appendix A shows the children’s breastfeeding, cereal introduction, and vaccine calendar data according to whether their mothers belonged to the intervention (IGC) or control group (CGC).

The children’s data according to maternal glucose tolerance status during pregnancy show no significant differences regardless of whether the mothers had been diagnosed with GDM or not. The results show a downward trend in the rates of atopic eczema, food allergies, and pharmacological treatments in the IGC, which did not reach statistical significance. These data are shown in Table 3.

Appendix A shows the children’s breastfeeding, cereal introduction, and vac-cine calendar data according to their mothers’ glucose tolerance status (GDM or normal glucose tolerance (NGT)).

When children were evaluated according to maternal adherence to nutritional recommendations during pregnancy (achieving a MEDAS score of at least 6 points), children whose mothers had a MEDAS score ≥ 6 showed lower rates of bronchiolitis [222 (28.8)/441(39.6), *p* = 0.022] and atopic eczema [311 (39.0)/564 (50.7), *p* = 0.039] than children whose mothers had a MEDAS score < 6. Likewise, a significantly smaller amount of the former children required hospital admissions, and the surgical LOSs were shorter. A tendency towards lower rates of asthma and food allergy episodes was observed in the IGC. This information is displayed in Table 4.

Appendix A show the children’s breastfeeding, cereal introduction, and vaccine calendar data according to the mother’s compliance to nutritional recommendations during pregnancy, in terms of the modified MEDAS > 6.

The children of the mothers categorized according to the MEDAS score during gestation in four categories—0–3 as reference, 4.5, 6–7, and 8–12—shown a progressive reduction in bronchiolitis and atopic dermatitis relative risk, which became statistically significant when ≥8 points were reached with rates (95% CI) of 0.75 (0.58–0.97) and 0.82 (0.69–0.98), respectively. The data can be seen in Table 5.

## 4. Discussion

The data provided in this study show that a higher degree of maternal adherence to the MedDiet during pregnancy is associated with a reduction in the rates of bronchiolitis and atopic dermatitis in offspring. Furthermore, the IGC had fewer episodes that required pharmacological treatment as well as a shorter hospital LOS following medical admissions. These results may suggest that the quality of maternal diet early in pregnancy influences not only the child’s susceptibility to infection or autoimmunity, but also the severity of clinical conditions if they develop.

Different studies have evaluated the effect of a MedDiet on the health of offspring. Unlike our study, some analyze what the mother ate retrospectively [1,2,3,26], and most are observational or cohort follow-up studies [27]. The results they found are contradictory [28,29,30]. The current study not only uses information on the maternal diet collected during the pregnancy, as opposed to retroactively, but also evaluates the influence of possible confounding factors such as both exclusive and complementary breastfeeding and food introduction. Both bias and the mandatory vaccination schedule were completed in both groups in a similar way, suggesting that they did not affect these results. On the other hand, the IGC who attended nursery were more numerous and started at a younger age, thus placing IGC at a greater risk of susceptibility to infection.

The current study also attempts to elucidate whether there is an optimal cut-off point in the MEDAS score that translates into health protection in the offspring. Taking a MEDAS score of 3 or less as the reference category, our study finds that any increase in the MEDAS score is associated with a reduction in the rates of both bronchiolitis and atopic dermatitis. Thus, those with high adherence, reaching at least a score of 8 points, presented a 25% and 18% reduction in the rates of bronchiolitis and atopic dermatitis, respectively.

The effect of a maternal MedDiet on children’s wheezing and/or atopic dermatitis has been examined in recent articles emphasizing different eating patterns and types of foods [3,17,31,32,33,34]. In one such publication [27], the authors evaluate scores of 3 or less vs. higher ones as regards respiratory function tests in children at 7 years of age. This study finds an improvement in respiratory function for each point increase in the MEDAS starting from 4 points.

Of note is the fact that several studies place special emphasis on the intake of olive oil. In a retrospective study [32], authors found that only maternal olive oil intake was associated with reduced wheezing, and not the MedDiet in and of itself, underlying the importance of this food. EVOO is known to be a rich source of monounsaturated fatty acids and has been associated with an improved inflammatory profile [12]. Nuts are rich in unsaturated fatty acids and other phytochemical constituents that also exert potential beneficial effects on inflammatory profiles and immunity [13]. However, two recent reviews associate the consumption of EVOO with a greater adherence to the MedDiet [35,36]; these results that coincide with our findings.

The health benefits for offspring are more marked in our current study than when the same children were studied at 2 years of age [17]. The difference between the findings at 2 and 6 years could be related, at least in part, to the immune protection conferred to the younger children by breastfeeding, and immune stimulation by vaccination and daycare, thereby masking the beneficial effect of the maternal gestational diet. In fact, during the first 2 years of life, breastfeeding could exert a greater influence on the child’s health than the mother’s diet during pregnancy. To evaluate this latter influence, more follow-up time is required, with children 6 years old or older studied.

The beneficial effects seen in the IGC may also be linked to a reduction in the rate of GDM. In fact, it is known that GDM adversely affects the health of offspring [9,10]. Yet, when the data are analyzed according to glucose tolerance status during pregnancy, no significant differences are found in the rates of these diseases. It should be noted that women with GDM received the same nutritional treatment from 24 to 28 GW (screening of GDM) through delivery, even if they had initially been in the CG, increasing their consumption of EVOO and nuts, as had IG women from 12–14 GW onwards. We have previously described that the medical nutritional treatment based on the MedDiet of GDM induces a rate of adverse events during pregnancy similar to non-diabetic events [37,38,39]. Additionally, this MedDiet intervention may induce structural and functional changes in the infant’s organ systems during fetal development [11,40,41,42] and favorable epigenetic changes.

### 4.1. Clinical Implications

The findings of this study have important clinical implications. First, they reinforce the need to promote reinforced MedDiet strategies from the beginning of pregnancy, not only to prevent maternal and neonatal complications, but also to improve long-term health outcomes in the offspring. Second, this study also attempts to discern whether there is an optimal cut-off point related with improved offspring health. With a reference score ≤ 3, our study finds that any increase in the MEDAS score is associated with a reduction in the rates of both bronchiolitis and atopic dermatitis. In the children of mothers with a high dietary adherence, reaching at least 8 points, a 25% and 18% reduction in the rates of bronchiolitis and atopic dermatitis were observed, respectively. This may have practical significance, as it suggests that the goal in women from the onset of pregnancy should be a MEDAS score of at least 8 points for improved health outcomes in offspring.

### 4.2. Limitations

This study has several limitations. First, this study compares two types of MedDiet, one supplemented with EVOO and nuts, and another with the restriction of both. There was only a 2-point difference in the MEDAS score of intervention and control group women. Yet, this has been enough to detect differences in the rates of bronchiolitis and atopy, although this is probably insufficient to demonstrate differences in other conditions, such as asthma and food allergies. A second limitation is the fact that questionnaires that ask about specific symptoms can cause rates to be overestimated [43], although these data in our study have been confirmed in the pediatric history collected in the electronic registry.

### 4.3. Strengths

Among the main strengths of the present study are its prospective interventional randomized clinical trial design, the large sample size, the high follow-up rate of participants (over 80% at six years postpartum), and the use of electronic medical records to validate reported medical events. Our study also evaluates the main known confounding factors that can affect the interpretation of the results.

### 4.4. Future Research Directions

In summary, small increases in MEDAS score are associated with a lower rate of bronchiolitis and atopy at the age of 6 years and a non-significant trend of reduction in the rate of other diseases. These findings suggest that there could be additional benefits to the health of the offspring as regards other infectious and autoimmune diseases at older ages, for example, between 10 and 12 years of age, as well as in other metabolic diseases including overweight/obesity. Considering the high rate of postnatal follow-up, an evaluation of the offspring at 12 years of age is being planned.

## 5. Conclusions

In summary, the most relevant aspect of our study is that it represents what the real dietary pattern is in a Mediterranean country. Not only should EVOO and nut consumption not be restricted during pregnancy, but their intake is associated with a higher adherence to the Mediterranean pattern, with health benefits for the offspring to at least 6 years of age. This has practical significance. A high MEDAS score, ≥8, should therefore be a goal in the dietary management of pregnant women from early on in gestation, given the short- and long-term benefits. For this reason, we have developed a program that makes nutritional information available to all pregnant women.

## Figures and Tables

**Figure 1 nutrients-17-01719-f001:**
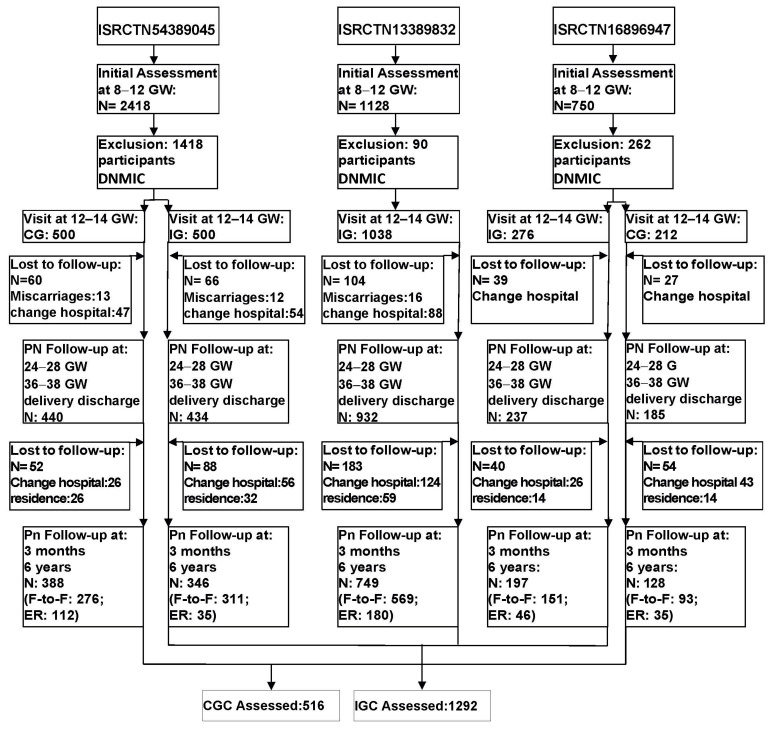
Flowchart of the St. Carlos cohort included in the analyses. GW: gestational week; IG: intervention group; CG: control group; PN: prenatal; DNMIC: did not meet inclusion criteria; Pn: postnatal; F-To-F: face-to-face; ER: electronic record; CGC: children from the control group; IGC: children from the intervention group.

**Table 1 nutrients-17-01719-t001:** Baseline characteristics of mothers whose children were analyzed at 6 years of age.

	Control Group *n* = 516	Intervention Group *n* = 1292	SS
Age (years)	32.0 ± 5.6	33.2 ± 5.0	0.001
Race/Ethnicity			
Caucasian/Iberian	304 (59.6)	851 (66.0)	0.027
Latin American	189 (36.5)	405 (31.4)	
Others	23 (3.9)	33 (2.6)	
Family history of Type 2 Diabetes	114 (22.1)	360 (27.9)	0.072
MetS (>2 components)	97 (18.8)	260 (20.2)	
Previous history of			
Gestational DM	15 (2.9)	56 (4.5)	0.022
Miscarriages	176 (34.1)	457 (35.4)	
Educational status			
Elementary education	60 (11.7)	75 (5.8)	0.007
Secondary School	156 (30.3)	329 (25.5)	
University Degree	292 (56.7)	869 (67.4)	
UNK	8 (1.4)	17 (1.2)	
Employed	391 (75.9)	1048 (81.2)	0.030
Number of pregnancies			
Primiparous	198 (38.4)	535 (41.6)	0.789
Second pregnancy	161 (31.2)	395 (30.7)	
>2 pregnancies	157 (30.4)	355 (27.6)	
Smoker			
Never	298 (57.8)	751 (58.2)	0.903
Current	39 (7.6)	105 (8.1)	
Gestational Age (weeks) at baseline	12.1 ± 0.6	12.0 ± 0.3	0.899
Pre-pregnancy Body Weight (kg)	62.2 ± 11.8	61.3 ± 10.7	0.070
Pre-pregnancy BMI (kg/m^2^)	23.8 ± 4.2	23.2 ± 3.8	0.007
BMI ≥ 25 kg/m^2^	162 (31.4)	314 (24.3)	0.801
MEDAS Score			
12 GW	4.7 ± 1.7	4.7 ± 1.7	0.242
24 GW	4.6 ± 1.8	5.8 ± 1.8	0.001
36 GW	5.6 ± 1.8	6.3 ± 1.9	0.001
Nutrition Score			
12 GW	0.2 ± 3.3	0.4 ± 3.1	0.098
24 GW	1.0 ± 3.4	3.1 ± 3.3	0.001
36 GW	3.0 ± 3.8	4.4 ± 3.7	0.001
Physical Activity Score			
12 GW	−1.7 ± 1.0	−1.9 ± 1.0	0.036
24 GW	−1.8 ± 1.0	−1.8 ± 0.9	0.691
36 GW	−1.8 ± 0.7	−1.6 ± 0.9	0.010

Data are Mean ± SD or number (%). MetS, Metabolic Syndrome. UNK, unknown. BMI, body mass index; GW, gestational weeks; DM, Diabetes Mellitus; MEDAS Score, the modified 12-point Mediterranean Diet Adherence Screener (MEDAS). Physical Activity Score, (walking daily (>5 days/week) Score 0: At least 30 min. Score +1, if >60 min. Score −1, if <30 min. Climbing stairs (floors/day, >5 days a week): Score 0, between 4 and 16; Score +1, >16; Score −1: <4).

**Table 2 nutrients-17-01719-t002:** Children’s data at 6 years of age according to whether their mothers belonged to the intervention (IGC) or control group CGC).

	CGC	IGC	*p*
Number	516	1292	
Age at 6 years follow-up (months)	70.3 ± 4.3	70.6 ± 4.4	0.139
Body weight (Kg)	23.0 ± 4.8	22.2 ± 4.1	0.002
Z score	0.164 ± 1.17	0.014 ± 1.07	0.017
Height (cm)	118.4 ± 5.8	117.9 ± 5.5	0.160
Z score	0.178 ± 1.15	0.159 ± 1.18	0.774
BMI (kg·m^−2^)	16.4 ± 2.6	15.9 ± 2.2	0.001
Z score	0.116 ± 1.18	−0.089 ± 1.04	0.002
Months achieving			
Anal sphincter control	31.6 ± 1.9	32.0 ± 13.8	0.555
Bladder sphincter daytime control	31.3 ± 9.9	31.5 ± 9.9	0.149
Bladder sphincter night control	36.7 ± 15.2	36.0 ± 26.0	0.605
Point out action	15.9 ± 4.4	15.5 ± 4.4	0.492
Walking: first steps	13.5 ± 2.9	12.8 ± 2.5	0.001
Diagnoses			
Food allergies	36 (7.6)	80 (6.2)	0.248
Asthma	23 (4.6)	40 (3.1)	0.083
Bronchiolitis	271 (52.5)	392 (30.4)	0.006
Atopic dermatitis	341 (69.0)	534 (41.3)	0.005
Pharmacological treament	397 (76.9)	892 (69.1)	0.001
Treatment with antibiotics	317 (66.5)	885 (68.5)	0.456
Number of episodes	2.5 ± 1.8	2.4 ± 1.8	0.505
Treatment with corticosteroids	259 (50.1)	676 (51.4)	0.327
Number of episodes	2.5 ± 2.5	2.4 ± 2.6	0.586
Hospitalization for severe disease:			
All-cause hospital stays	59 (11.4)	121 (9.4)	0.108
Children	54 (14.9)	46 (13.6)	0.079
Surgical	30 (5.8)	51 (3.9)	0.600
LOS (days)	4.8 ± 1.2	4.7 ± 1.3	0.157
Medical	33 (6.4)	81 (6.3)	0.408
LOS (days)	8.3 ± 6.0	4.8 ± 4.4	0.048

Results expressed as mean ± SD or *n* (%). BMI, body mass index; LOS, length of stay; CGC, control group children; IGC, intervention group children.

**Table 3 nutrients-17-01719-t003:** Children’s data at 6-year follow-up according to their mothers’ glucose tolerance (GDM or NGT).

	GDM	NGT	*p*
Number	320	1488	
Gestation age at delivery	39.4 ±1.7	39.5 ± 1.5	0.164
Months achieving			
Anal sphincter control	31.6 ± 10.0	32.2 ± 14.9	0.635
Bladder sphincter daytime control	31.9 ± 10.4	31.5 ± 9.9	0.591
Bladder sphincter night control	35.1 ±12.4	36.2 ± 28.7	0.605
Point out action	15.5 ± 4.4	15.5 ± 4.3	0.942
Walking: first steps	13.4 ± 3.0	13.4 ± 2.8	0.893
Diagnoses *n* (%)			
Food allergies	32 (7.8)	84 (5.7)	0.046
Asthma	16 (1.9)	47 (3.2)	0.107
Bronchiolitis	196 (29.9)	467 (31.4)	0.441
Atopic dermatitis	141 (43.9)	734 (49.3)	0.113
Pharmacological treament	257 (80.1)	1032 (69.4)	0.052
Treatment with antibiotics	216 (67.3)	986 (66.3)	0.267
Number of episodes	2.2 ± 1.5	2.4 ± 1.8	0.176
Treatment with corticosteroids	199 (49.2)	736 (49.5)	0.164
Number of episodes	2.5 ± 2.1	2.5 ± 2.7	0.913
Hospitalization for severe disease:			
All-cause hospital stays	26 (8.1)	154 (10.3)	0.136
Children	16 (5.2)	84 (5.6)	0.079
Surgical	17 (5.3)	64 (4.3)	0.681
LOS (days)	4.3 ± 2.6	4.1 ± 2.3	0.845
Medical	22 (6.9)	92 (6.2)	0.364
LOS (days)	4.9 ± 4.6	4.8 ± 4.0	0.438

Results expressed as mean ± SD or *n* (%). GDM, Gestational Diabetes Mellitus. NGT, normal glucose tolerance; LOS, length of stay.

**Table 4 nutrients-17-01719-t004:** Children’s data at 6 years of age according to mother’s compliance to nutritional recommendations during pregnancy in terms of modified Mediterranean Diet Adherence Screener Score ≥ 6 (MEDAS).

MEDAS	≥6	<6	*p*
Number (*n*)	695	1113	
Gestation age at delivery	39.6 ± 1.5	39.5 ± 1.6	0.195
Months achieving			
Anal sphincter control	31.7 ± 10.8	32.0 ± 10.0	0.597
Bladder sphincter daytime control	31.0 ± 10.0	31.8 ± 10.0	0.061
Bladder sphincter night control	36.7 ±12.9	36.0 ± 14.1	0.099
Point out action	15.6 ± 4.6	15.7 ± 4.3	0.828
Walking: first steps	13.3 ± 2.8	13.3 ± 2.8	0.879
Diagnoses			
Food allergies	45 (6.5)	71 (6.4)	0.474
Asthma	24 (3.5)	39 (3.5)	0.253
Bronchiolitis	222 (28.8)	441 (39.6)	0.022
Atopic dermatitis	311 (39.0)	564 (50.7)	0.039
Pharmacological treament	371 (53.4)	918 (82.5)	0.480
Treatment with antibiotics	425 (67.6)	777 (69.9)	0.077
Number of episodes	2.3 ± 1.7	2.5 ± 1.9	0.060
Treatment with corticosteroids	327 (52.9)	608 (54.6)	0.089
Number of episodes	2.4 ± 2.3	2.6 ± 2.7	0.084
Hospitalization for severe disease			
All-cause hospital stays	65 (9.4)	115 (10.3)	0.394
Children	26 (3.7)	74 (6.7)	0.039
Surgical	25 (3.6)	56 (5.0)	0.087
LOS (days)	3.6 ± 2.6	4.6 ± 2.3	0.030
Medical	43 (6.2)	71 (6.4)	0.452
LOS (days)	3.2 ± 3.3	3.2 ± 3.6	0.686

Results expressed as mean ± SD or *n* (%). LOS, length of stay.

**Table 5 nutrients-17-01719-t005:** Children’s data at 6 years of age according to the maternal MEDAS score during pregnancy.

	Cases	MEDAS 0–3	SCORE 4–5	6–7	8–12	Trends *p*
*N*	1808	564	549	519	176	
Food allergies	116	32 (5.7)	39 (7.1)	33 (6.4)	12 (6.8)	0.080
RR (95% CI)		Reference	1.14 (0.92–1.41)	1.03 (0.80–1.32)	1.61 (0.98–2.37)	
Asthma	63	23 (4.1)	16 (2.9)	17 (3.3)	7 (3.9)	0.054
RR (95% CI)		Reference	0.85 (0.56–1.29)	0.98 (0.68–1.41)	0.91 (0.70–1.17)	
Bronchiolitis	663	246 (43.6)	195 (35.5)	166 (32.0)	56 (31.8)	0.029
RR (95% CI)		Reference	0.99 (0.77–1.30)	0.82 (0.57–1.19)	0.75 (0.58–0.97)	
Atopic dermatitis	875	296 (52.5)	268 (48.8)	243 (46.8)	68 (38.6)	0.039
RR (95% CI)		Reference	0.99 (0.88–1.12)	0.94 (0.83–1.07)	0.82 (0.69–0.98)	
Pharmacological T.	1289	458 (81.5)	460 (83.8)	248 (47.8)	123 (69.9)	0.480
RR (95% CI)		Reference	0.95 (0.70–1.28)	0.83 (0.68–1.06)	0.96 (0.69–1.33)	
With antibiotics	1202	355 (62.9)	422 (76.9)	354 (68.2)	71 (40.3)	0.077
RR (95% CI)		Reference	1.03 (0.90–1.17)	1.05 (0.81–1.36)	0.89 (0.68–1.17)	
With corticosteroids	935	319 (56.6)	289 (52.6)	275 (53.0)	52 (29.5)	0.089
RR (95% CI)		Reference	0.87 (0.62–1.22)	1.09 (0.96–1.22)	0.84 (0.66–1.06)	
Hospitalization for severe disease						
All-cause hospital stays	180	60 (10.6)	55 (10.0)	52 (10.0)	13 (7.4)	0.541
RR (95% CI)		Reference	0.85 (0.58–1.25)	0.89 (0.60–1.33)	0.81 (0.49–1.35)	
Children	100	18 (3.2)	56 (10.2)	17 (3.3)	9 (5.1)	0.139
Surgical	81	13 (2.3)	43 (7.8)	19 (3.7)	6 (3.4)	0.541
Medical	114	41 (7.3)	30 (5.5)	35 (6.7)	8 (4.5)	0.454

Results expressed as number of cases (%) and relative risk RR and 95% CI (confidence interval). MEDAS, modified 12-point Mediterranean Diet Adherence Screener. T, treatment.

## Data Availability

The original contributions presented in the study are included in the article/Appendix A, further inquiries can be directed to the corresponding authors.

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
