# Peer review of "A Greater Adherence to the Mediterranean Diet Supplemented with Extra Virgin Olive Oil and Nuts During Pregnancy Is Associated with Improved Offspring Health at Six Years of Age"

_nutrients, 2025, doi:10.3390/nu17101719_

Round 1
Reviewer 1 Report
Comments and Suggestions for Authors
1) The is too long and should be shortened to appear clear and concise
2) The keywords should differs from those who appear in the title in order to increase the probability in find the paper once published
3) The introduction section is poorly described and referenced, should be more elaborated around the rational behind the conduction of the study
4) After the aim of the study a hypothesis should be stated
5) Due to the large number of variables included in the study an adjustment of the level of significance is needed (i.e. Bonferroni), even if there is no multi-comparison analysis
6) The discussion section is highly speculative, authors are kindly requested to structure it in subsections according to the following titles:
- The main findings of the study and their comparison with previously published literature on a national and international level
- The clinical implication of the study
- The strength and limitations of the study
- The new directions for the future research
7) Authors should consider the improvement of tables, by reducing there total number as well as the number of variables in each included
8) The reference section should be enriched much more
Author Response
Thank you very much for your constructive comments with which we agree. We have to make the following considerations:
Comments and Suggestions for Authors
- The is too long and should be shortened to appear clear and concise.
R: According to your comments, we have simplified the tables, reduced the number of rows in the tables in the text, and provided as a supplementary table the analysed data of the variables not related to the intervention but that can influence the results (bias)
- The keywords should differ from those who appear in the title in order to increase the probability in find the paper once published
R: According to your comments we have introduced different keywords from those included in the title
- The introduction section is poorly described and referenced, should be more elaborated around the rationale behind the conduction of the study
R: We have expanded the introduction and referenced, many used in the discussion and we have adapted the paragraph of the objective and hypothesis to be tested
4) After the aim of the study a hypothesis should be stated
R: we have adapted the paragraph of the objective and hypothesis to be tested
5) Due to the large number of variables included in the study an adjustment of the level of significance is needed (i.e. Bonferroni), even if there is no multi-comparison analysis
R: As the variables related to the intervention, adherence to the Mediterranean diet and glucose tolerance during pregnancy, are the most important variables related to the health of the offspring, we have preferred to provide RR data (95%CI) categorized by MEDAS score during pregnancy in 2 ways, above and below the mean, and in 4 levels 0-3, which is taken as a reference, 4-5. 6-7 and 8-12, as well as normal glucose tolerance or GDM. We think it's more informative. Confounding variables that could affect the results do not have significant differences. Although we adjust for age (continuous) parity (primiparous vs. multiparous categorical) the RRs are not modified and therefore we do not provide it so as not to increase the data provided. The adjustment for BMI (categorical <15, 25-29.9 and 30 or more) should be considered inadequate since the third RCT only includes overweight women
6) The discussion section is highly speculative; authors are kindly requested to structure it in subsections according to the following titles:
- The main findings of the study and their comparison with previously published literature on a national and international level
- The clinical implication of the study
- The strength and limitations of the study
- The new directions for the future research
R: We have re-ordered the discussion following their recommendations
7) Authors should consider the improvement of tables, by reducing there total number as well as the number of variables in each included
R: As we have said before, we have re-adapted the content of the tables
8) The reference section should be enriched much more
R: We have added relevant references

Reviewer 2 Report
Comments and Suggestions for Authors
The work presented in this manuscript is a cohort study for investigating effect of slightly different MedDiet of mother during pregnancy on the health of offspring at 6 years of age in Spain.
This study found more health benefit (e.g., lower prevalence of bronchiolitis and atopic eczema) in children of mothers with MedDiet supplemented with olive oil and pistachios when compared with children of mothers without oil and nuts supplementation.
Effect of maternal diet on health of the offspring is a crucial issue and this study is interesting and valuable in this respect.
The following points are to be considered.
- The study cohort is composed of 3 different populations with slightly different design in terms of dietary treatment and subjects’ characteristics. The authors are required to justify the inclusion of subjects from the 3 populations into the same category (IG and CG) for the statistical analyses.
- Control group in this study is not the control of intervention group in exact meaning: the same basic MedDiet with urged lower fat intake than IG. Theoretically speaking, the health benefit observed among IG could be partly due to greater fat intake than CG but the authors did not refer to this in this manuscript.
- How valid the supplemented olive oil and nuts were actually consumed by the IG mothers? How did the authors confirmed the validity?
- Validity of the data used in this study, e.g., children’s health conditions, must be referred.
- The authors analyzed the data with simple univariate statistical analyses. I’d like to know the reason why the authors did not try multivariate regression analysis in this study.
- Line 198-200. The description “nonparametric tests have been used for continuous variables with normal distribution” must be a mistake.
Author Response
Thank you very much for your constructive comments with which we agree. We have to make the following considerations:
Comments and Suggestions for Authors
The work presented in this manuscript is a cohort study for investigating effect of slightly different MedDiet of mother during pregnancy on the health of offspring at 6 years of age in Spain.
This study found more health benefit (e.g., lower prevalence of bronchiolitis and atopic eczema) in children of mothers with MedDiet supplemented with olive oil and pistachios when compared with children of mothers without oil and nuts supplementation.
Effect of maternal diet on health of the offspring is a crucial issue and this study is interesting and valuable in this respect.
The following points are to be considered.
- The study cohort is composed of 3 different populations with slightly different design in terms of dietary treatment and subjects’ characteristics. The authors are required to justify the inclusion of subjects from the 3 populations into the same category (IG and CG) for the statistical analyses.
R: As we included in the experimental design, the nutritional intervention was the same in the 2 IGs of the studies and the CG recommendations in the 2 RCTs were identical. The third study was the universal application of the IG recommendation in the usual clinical practice in all pregnant women. We believe that it is adequately referred to in methods. We clarify
- Control group in this study is not the control of intervention group in exact meaning: the same basic MedDiet with urged lower fat intake than IG. Theoretically speaking, the health benefit observed among IG could be partly due to greater fat intake than CG but the authors did not refer to this in this manuscript.
R: As we have mentioned in the previous point, and as suggested, the results obtained may be due to a high consumption of fat in the GI in relation to the GC. However, this is associated with a higher MEDAS score during pregnancy, i.e. due to greater adherence to the MedDiet. For this reason, we specifically include tables 4 and 5.
- How valid the supplemented olive oil and nuts were actually consumed by the IG mothers? How did the authors confirmed the validity?
R: The validation of the consumption of EVOO and nuts was carried out by applying food frequency questionnaires (MEDAS and Nutrition). In the first RCT, biomarkers were also determined to validate their consumption: the levels of tocopherol and hydroxytyrosol that demonstrated a higher consumption in the GI compared to the GL between 24 and 12 GW
- Validity of the data used in this study, e.g., children’s health conditions, must be referred.
R: The child health data provided by the mother had to be contained in the health card of the offspring, and if there were diagnoses it had to be recorded in the electronic history of the outpatient paediatrician and in the hospital, discharge reports and in the pharmacy drug supply record (MUP). We believe that it is adequately referred to in methods
- The authors analysed the data with simple univariate statistical analyses. I’d like to know the reason why the authors did not try multivariate regression analysis in this study.
R: We preferred to provide RR data (95%CI) categorized by MEDAS score during pregnancy in 2 ways, above and below the mean, and at 4 levels 0-3, which is taken as a reference, 4-5, 6-7 and 8-12, as well as for normal glucose tolerance or GDM, We believe that it is more informative When we have adjusted for age and parity (the narrow range in both variables must be considered) the 95%CI and RR have not been modified
- Line 198-200. The description “nonparametric tests have been used for continuous variables with normal distribution” must be a mistake.
R: Indeed, it is an error. --- non-normal distribution

Round 2
Reviewer 1 Report
Comments and Suggestions for Authors.